# Investigating the Impact of Tillage and Crop Rotation on the Prevalence of *phlD*-Carrying *Pseudomonas* Potentially Involved in Disease Suppression

**DOI:** 10.3390/microorganisms11102459

**Published:** 2023-09-30

**Authors:** Ridhdhi Rathore, Dermot Forristal, John Spink, David Dowling, Kieran J. Germaine

**Affiliations:** 1EnviroCore, Dargan Research Centre, South East Technological University (SETU), R93 V960 Carlow, Ireland; ridhdhi.rathore@teagasc.ie (R.R.); david.dowling@setu.ie (D.D.); 2Teagasc Agriculture and Food Development Authority, Oak Park Research Centre, R93 XE12 Carlow, Ireland; dermot.forristal@teagasc.ie (D.F.);

**Keywords:** biocontrol, 2,4-DAPG, tillage, crop rotation, *Pseudomonas*

## Abstract

Winter oilseed rape (OSR) is becoming an increasingly popular crop in rotations as it provides a cash crop and reduces the incidence of take-all fungal disease (caused by *Gaeumannomyces graminis*) in subsequent wheat production. The exact mechanism of this inhibition of fungal pathogens is not fully understood; however, the selective recruitment of bacterial groups with the ability to suppress pathogen growth and reproduction is thought to play a role. Here we examine the effect of tillage practice on the proliferation of microbes that possess the *phl*D gene involved in the production of the antifungal compound 2,4-diacetylphloroglucinol (2,4-DAPG), in the rhizospheres of both winter oilseed rape and winter wheat grown in rotation over a two-year period. The results showed that conservation strip tillage led to a significantly greater *phl*D gene copy number, both in the soil and in the roots, of oilseed rape and wheat crops, whereas crop rotation of oilseed rape and wheat did not increase the *phl*D gene copy number in winter wheat.

## 1. Introduction

Micro-organisms with the ability to suppress the growth and spread of plant pathogens can be an effective alternative to traditional chemical-based control agents used in agriculture. Among such biocontrol agents, plant-growth-promoting (PGP) bacteria are an important group that help improve plant health and the suppression of pests and diseases. A meta-analysis study carried out by Lamichhane et al. [1], comprising 396 studies worldwide, revealed for the first time that seed treatment with PGP bacteria significantly improves seed germination (7 ± 6%), seedling emergence (91 ± 5%), plant biomass (53 ± 5%), disease control (55 ± 1%) and crop yield (21 ± 2%) compared to untreated seeds. The abundance and diversity of these bacteria may fluctuate according to soil and crop management practices such as tillage and crop rotation [2], crop species and variety [3], soil location [4] and soil geomorphology [5]. Additionally, the growing season of the crop is also a driver of bacterial population size in agricultural systems. Root system development over the growing season and associated changes in rhizodeposition may alter the spatial distribution and quality of organic materials [6], influencing the dynamics of the microbial community over time.

The bacterial genus *Pseudomonas* has many members possessing PGP abilities [7]. Their presence and abundance in the rhizosphere, rhizoplane and within the root can significantly improve plant health [8] and crop yields and/or help with disease management in agronomy [9,10]. Strains of *Pseudomonas* spp. can protect plants from fungal diseases through several different mechanisms, most notably, through the production of antimicrobial compounds such as 2,4-diacetylphloroglucinol (2,4-DAPG), and phenazine derivatives, pyrrolnitrin and pyoluteorin [11,12]. 2,4-DAPG is a polyketide antibiotic that has been reported to have antibacterial, antiviral, antifungal, antihelminthic and phytotoxic properties [13]. 2,4-DAPG can trigger induced systemic resistance (ISR) responses in plants leading to enhanced plant protection from phytopathogens [14]. 2,4-DAPG has a key role in biological control of plant diseases such as take-all in wheat (caused by *Gaeumannomyces graminis var. tritici*), black root rot (*Thielaviopsis basicola*) and Granville wilt (*Ralstonia solanacearum*) of tobacco and damping-off of sugar beet (*Pythium ultimum*) [15]. The genes involved in the biosynthesis of 2,4-DAPG are located on an 8-kb *phl* cluster and consists of eight genes; *phl*HGFACBDE. The cluster is conserved at the organizational level in 2,4-DAPG producing bacteria [16] and includes genes coding for biosynthetic and degradation enzymes (*phl*ACBD and *phl*G), transporters (*phl*E and *phl*I) and regulators (*phl*F and *phl*H). The production of 2,4-DAPG is strictly regulated by the GacS-GacA system, a global signal transduction system present in many Gram-negative bacteria. In plant- and animal-pathogenic bacteria, the GacS-GacA system is critical for virulence, whereas the same system is necessary to produce antibiotic compounds in plant-beneficial fluorescent pseudomonads [17]. 2,4-DAPG is transported out of the cell and into the soil. Okubara et al. [18] inoculated different wheat varieties with 2,4 DAPG-producing *Pseudomonas* strains. After 4 days, the levels of 2,4-DAPG in the rhizoplane were between 1650 and 2767 ng g^−1^, while Kwak et al. [19] estimated the half-life of 2,4-DAPG in the rhizosphere of wheat to be 0.25 days.

The production of 2,4-DAPG is limited to just a few groups of bacteria including Pseudomonad species such as *P. fluoresecens*, *P. chloraphis*, *P. brassicaerum* and *P. protegen* [20]. The population sizes of these *phl*+ *Pseudomonas* in the bulk soil and the rhizosphere can be high enough for effective plant protection. The effectiveness of *phl*+ strains can vary among different bacterial isolates and species. The key biosynthetic gene is *phlD* [21], which is required for the synthesis of phloroglucinol, a precursor of monoacetylphloroglucinol (MAPG) and 2,4-DAPG. *Pseudomonas* bacteria harbouring the *phlD* gene are found in soils worldwide [13,16]. Genetic variation in the *phl*D gene can affect the level of 2,4-DAPG production and, consequently, the strain’s ability to suppress pathogens and promote plant growth. In addition, their population size and diversity may fluctuate according to soil location, soil geomorphology, crop species and variety, and soil management. *Phl*D+ counts in vineyard soil were found to range from log 6.0 to log 8.0 but decreased with depth by 1.5–2 log [22]. Almario et al. [23] showed *phl*D^+^ population sizes of log 5–log 8 in the rhizosphere of tobacco plants grown in two *Thielaviopsis basicola* suppressive soils and two conducive soils and detected four genotypes of the *phlD* gene. Suresh et al. [24] isolated a total of 87 fluorescent *Pseudomonas* strains from the rhizosphere of tomato plants. In total, 35% of these exhibited antagonistic activity against the fungal pathogen *R. solanacearum*, with 2,4-DAPG production the most common biocontrol agent produced by them (11.5%). Burrow et al. [25] showed that 2,4-DAPG-producing *P. protegens*, when combined with the fungicide fluquinconazole, increased yield, grain quality, test weight parameters, biomass and plant height at the early stages of plant development and decreased the severity of disease. Their results suggest that the integration of *Pseudomonas* strains with fluquinconazole is an alternative management strategy for reducing take-all disease in southern Chile. Patel and Archana [26] created a genetically engineered PGPR *Pseudomonas* strain with enhanced production of 2,4-DAPG. The strain could protect inoculated plants from infection with *Magnaporthe oryzae* and *Rhizoctonia solani*.

Conservation practices, such as strip tillage (ST), have been widely used to counteract the negative effects of farming practices used in intensive conventional tillage (CT) systems. Conservation strip tillage practices are recognized for their advantages in reducing input costs, enhancing water use efficiency and preserving soil carbon. Li et al. [27] carried out a global meta-analysis of conservational tillage research and showed that conservation tillage practices lead to statistically significant positive effects on many soil physical properties. However, these studies did not examine the effect of conservation tillage practices on the populations of beneficial soil microbes, and so this remains a significant knowledge gap. While many research studies have reported the abundance of *Pseudomonas* bacteria that harbour the genes involved in the biosynthesis of antimicrobial compounds in the bulk soil, rhizosphere and root interior [28,29,30], research investigating the combined impact of tillage practices and crop rotation on population density is limited. Therefore, the work detailed in this study focused on understanding the contribution of tillage practices and crop rotation together on the prevalence of the *phlD* gene in the rhizosphere and roots of *Brassica napus* (oil seed rape (OSR)) and *Triticum aestivum* (wheat) (in rotation and in monoculture) crops at different plant growth stages, over two growing seasons using quantitative real-time PCR (qPCR). We hypothesized that the crop rotation of wheat with OSR would lead to significant differences in the prevalence of *phl*D+ bacteria between conventional tillage (CT) and conservation strip tillage (ST) practices throughout the plant cultivation cycle.

## 2. Materials and Methods

### 2.1. Experimental Design

The plant samples for this study were collected from a field experiment evaluating the impact of crop rotation along with crop establishment systems on the growth, development and production of cereal and break crops. The establishment systems comprised of (1) conventional tillage (CT): a conventional plough-based system, and (2) strip tillage (ST): a low-disturbance conservation strip tillage. These trials have been conducted continuously since the year 2012, where five different winter crops have been grown in rotation; for example, OSR followed by wheat, oats, wheat, barley and again OSR under CT and ST tillage practices as described in Appendix A. The growth period of each winter crop from sowing to harvesting was ~300 days.

The trials were a randomized block design where the main plots comprised the tillage practices and subplots were divided into crop rotations. The individual plot’s dimensions were 24 m × 4.8 m. The conventional establishment system comprised of mouldboard ploughing, which inverted the soil to a depth of 230 mm, two days prior to sowing. The ploughed soil then received a secondary cultivation to 100 mm depth with a rotary power harrow. The OSR and wheat were sown at 10 mm depth at a row spacing of 125 mm using a conventional mechanical delivery seed drill operated in combination with the power harrow. The low-disturbance establishment system deployed was a non-inversion system, comprised of a single cultivation/seeding pass of a rigid leg cultivator, which were operated at a 200 mm depth. These forward-facing tines, with side ‘wings’ gave additional soil disturbance, worked directly in the cereal residue of the previous crop, disturbing approximately 50% of the surface width between the legs. Seeding was by metered pneumatic delivery of seeds to a point behind the cultivator leg, giving a row spacing of 600 mm in OSR and 330 mm in wheat crops. After seeding with both establishment systems, the soil surface was rolled using a ring roller. Crop management, other than crop establishment, followed standard practices for winter OSR and winter wheat production in this region, which are illustrated in Appendix A. The basic soil analysis is presented in Appendix A. 

### 2.2. Sample Collection and Treatment

For this study, plants were sampled from three different rotational systems: (1) OSR (previous crop was barley), (2) wheat (previous crop was OSR) and (3) from monoculture wheat (previous crop was wheat) under CT and ST tillage practices. Samples were collected in triplicate from each block of three replicated plots at three different growth stages: the vegetative stage (~120 days after sowing), flowering stage (~240 days after sowing) and at the harvesting (~330 days after sowing) stage in two sequential years, 2014-15 and 2015-16. Meteorological conditions during these years are presented in Appendix A.

The plant samples were processed into two compartments, i.e., rhizosphere soil and root. The excess soil from the root was removed by manual shaking, leaving ~1 mm of rhizosphere soil still attached to the root. The rhizosphere soil attached to the roots was scraped off with a sterile forceps and collected in prelabelled sterile 50 mL falcon tubes. The root samples were washed separately in sterile 50 mL Falcon tubes containing 30 mL of Phosphate-Buffered Saline (PBS) and washed for 20 min at 180 rpm on a shaking platform. The roots were transferred to a new Falcon tube and subjected to a second washing treatment (20 min at 180 rpm in 10 mL PBS buffer). Doubled-washed roots were then transferred to a new Falcon tube and sonicated for 10 min at 160 W in 10 intervals of 30 s pulse and 30 s pause (Bioruptor Next Gen UCD-300, diagenode, Liège, Belgium) to remove the tightly adhered microbes from the root surface. Roots were removed from PBS, rinsed in a fresh volume of 20 mL PBS buffer and ground with a mortar and pestle in liquid nitrogen. Pulverised roots were collected in 50 mL Falcon tubes and stored at −80 °C until further processing. All the samples were stored in −80 °C until required for the DNA extraction.

### 2.3. Bacterial Strain and Culture Storage

*Pseudomonas ogarae* F113 (formerly known as *Pseudomonas fluorescens* F113) [31], a known 2,4-DAPG producer, was used as a source of the *phl*D gene and as a positive control in the qPCR assays. The bacterial strain was cultured from −80 °C frozen stocks and regenerated in King’s B (KB) media. The plate was inverted and incubated for 24 h at 28 ± 2 °C. Single isolated colonies were removed from the plates and used to inoculate sterile KB broth. 

### 2.4. DNA Extraction and Determining Quality–Quantity of DNA

Genomic DNA from *P. ogarae* F113 was cultured overnight in KB medium. A Wizard^®^ Genomic DNA extraction kit (Promega, Madison, WI, USA) was used to extract DNA from the cultured cells following the manufacturer’s instructions, and the extracted DNA samples were stored at −20 °C. DNA extraction from the rhizosphere soil samples was performed using the MoBio PowerSoil^TM^ DNA isolation kit (Carlsbad, CA, USA) as per the manufacturer’s guidelines. Total soil DNA was eluted in 50 µL of sterile nuclease free water (Sigma Aldrich, Burlington, MA, USA). DNA extraction from plant root samples was performed on the ~0.5 g of plant tissues using the 2% acetyl trimethylammonium bromide (CTAB) method, and DNA was dissolved in 100 µL of sterile water. The concentration and purity of DNA was determined by Nanodrop spectrophotometry (Thermo Scientific, Wilmington, DE, USA). Post-quantification, all DNA samples were normalized to 10 ng µL^−1^. The three DNA samples from each microhabitat zone per block were pooled (e.g., the three DNA samples from the rhizosphere soil samples from block 1 were pooled) to give representative DNA samples of the rhizosphere and root from each block. 

### 2.5. Generation of Standard Curve and phlD Quantification in Samples

Quantitative PCR with SYBR green technology was used to quantify the *phl*D gene copy numbers from the root and rhizosphere soil samples of OSR and wheat plants. A standard curve of *phlD* copy number was generated using genomic DNA of the *phlD P. ogarae* F113 bacterial strain. Genomic DNA of this bacteria was serially diluted ten-fold in three separated series to obtain standards from 3 × 10^6^ to 30 fg DNA µL-1. One microliter of each standard dilution (i.e., from approximately 4 × 10^5^ to 4 *phl*D copies) was used for qPCR analysis. QPCR assays were conducted using 96-well white microplates, Roche SYBR green master mix in a final volume of 10 µL and a LightCycler 96 (Roche Applied Science, Meylan, France). 

The reaction mixture contained 5 µL of Roche SYBR Green I Master Vial 1, 3 µL of Vial 2 (Roche Applied Science), 0.5 µL of primer B2BF (1 µM), 0.5 µL of primer B2BR3 (1 µM) and 1 µL of DNA. The sequences of the primers used are detailed in Table 1. The final cycling program included a 10 min incubation at 95 °C, 50 amplification cycles of 30 s at 94 °C, 7 s at 67 °C and 15 s at 72 °C. Amplification specificity was checked by melting curve analysis of the amplification product using a fusion program consisting of an initial denaturing step of 5 s at 95 °C, an annealing step of 1 min at 65 °C and a denaturing temperature ramp from 65 to 97 °C with a rate of 0.11 °C s^−1^. The cycle threshold (Ct) of each individual sample was calculated using the second derivative maximum method in the LightCycler 96 software v 1.5 (Roche Applied Science). 

The standard curve was obtained by plotting the mean Ct value of the three replicates (per DNA concentration) against the log-transformed DNA concentration. Amplification efficiency (E), calculated as E = 10 (−1/slope) −1, and the error of the method (mean squared error of the standard curve) were determined using the LightCycler software v1.5 (Roche Applied Science). The equivalence between DNA amount and *phl*D copy number was estimated based on (i) a *Pseudomonas* genome of approximately 7.26 fg DNA and (ii) the occurrence of one *phl*D copy per genome. The detection limit was determined as the number of *phl*D copies giving 3 positive results out of 3 replicates. The amplification curve, melting curve and standard graph are displayed in Appendix A. Melting curve calculation and Tm determination were performed using the Tm Calling Analysis module of LightCycler 96 software v1.5 (Roche Applied Science). The standard curve thus generated from genomic DNA of *P. ogarae* F113 was subsequently used as the external standard curve for determination of the *phl*D copy number in DNA samples. The equimolar concentration (10 ng µL^−1^) of the DNA samples of root and rhizosphere soil were analysed by qPCR in triplicate (following the above protocol), and the mean Ct value was reported in the external standard curve to infer the *phl*D copy number in the sample, using the LightCycler 96 software and the ‘standard curve’ option for the absolute quantification. Positive control (*P. ogarae* F113 genomic DNA 30 ng µL^−1^), water control and three DNA standards from the genomic DNA of *P. ogarae* F113 (3000 pg, 30 pg and 0.3 pg corresponding to approximately 4 × 10^5^, 4 × 10^3^ and 40 copies) in triplicate were included as a reference in each run to detect between-run variations. 

### 2.6. Calculating Copy Number (CN) g^−1^ of Soil and g^−1^ of Root Samples following qPCR

Sample concentration was determined following the amplification cycle. The standard curve was developed to calculate the gene CN of a 10 μL reaction of each dilution. The Ct of the unknown sample was then extrapolated from the equation line of the standard curve. The concentration was then back calculated to determine the CN g^−1^ of the original soil or root sample. The concentration of each DNA sample (both soil and root) was normalized to 10 ng μL^−1^, and in each 10 μL reaction mixture, 1 μL normalised DNA was used. Examples of the calculations are shown in Appendix A. 

### 2.7. Statistical Analysis

Statistical analyses were performed in R v3.4.2 operated through R Studio v0.99.893. The average values obtained from the three technical replicates of each qPCR assay were used for statistical analysis. The normal distribution of data was checked with the Shapiro–Wilk test. Significant differences in the variance of parameters were assessed using the non-parametric Mann–Whitney–Wilcoxon and Kruskal–Wallis tests to identify significant differences between the two tillage practices, years and between the crops. Post hoc comparisons were conducted by the Kruskal–Wallis Dunn test. Data used for the heat map showing rankings of OSR and wheat crops with their respective microhabitat zones of *phl*D abundance at different growth stages under tillage practices were normalized using the function ‘scale’ (R package ‘stats’), and the graph was visualized using the package ‘heatmap. Plus’.

## 3. Results

The abundance of bacteria harbouring the *phl*D gene was quantified by qPCR in both the rhizosphere soil and root samples of winter OSR, and wheat crops grown in rotation and in continuous culture under conventional tillage (CT) and conservational strip tillage (ST) practices in two trials run over two consecutive years. To the best of our knowledge, there has only ever been one copy of the *phl*D gene per cell in the 2,4-DAPG strains isolated to date. As such, we assume that one *phl*D gene copy number represents one bacterial cell [28]. However, this may not be the case, and therefore, our results may overestimate the number of 2,4-DAPG-producing cells in our samples. The *phl*D gene was detected in 94% of samples in year 1 and 100% of samples in year 2. This highlights the almost ubiquitous nature of bacteria that carry the *phl*D gene in both the oilseed rape and wheat microbiomes in these trials and suggests that production of 2,4-DAPG is an important trait in these soil microbial communities.

### 3.1. Frequency of phlD in the Rhizospheres of OSR and Wheat Crops under Different Tillage Systems

*phl*D gene copy numbers were quantified at three plant growth stages (vegetative, flowering and harvesting) in both the rhizosphere and root tissues of OSR and wheat (in rotation and in monoculture), under convention tillage (CT) and strip tillage (ST) systems. The mean Ct value and calculated gene copy number per gram of each soil/root sample are presented in Appendix A. In the wheat rhizosphere, there was generally a decreasing abundance of the *phl*D gene over the growing cycle (~1 order of magnitude drop between each stage). However, in the OSR rhizosphere, there tended to be a drop in abundance at the flowering stage, but this recovered to even greater levels at the harvesting stage. This was observed in both year 1 and year 2.

At the vegetative stage, the copy numbers of the *phl*D gene in both the OSR and wheat rhizospheres were significantly higher (~1 to 1.5 log) in ST samples compared to CT samples (Figure 1A; *p* < 0.001). There was no significant difference in *phl*D abundance between OSR-CT plots and WCT plots, nor was there a significant difference between the OSR ST plots and the wheat ST plots (rotation or monoculture). A similar pattern of *phl*D abundance was recorded in both years as a result of crop rotation and tillage practice. There was no significant difference between the vegetative stage in year 1 and the vegetative stage in year 2 (*p* > 0.05).

At the flowering stage (Figure 1B), the *phl*D gene copy numbers in both the OSR and wheat rhizospheres were higher in ST samples compared to CT samples. This increase was statistically significant in 50% of the comparisons. In year-1, there was no statistical difference in the *phl*D gene in OSR compared to the two wheat trials. In year-2 samples, all the conventional tilled plots (CT) saw a significant decline in the *phl*D gene copy number (*p* < 0.05), whereas in ST samples, *phl*D abundance was relatively stable. Furthermore, there was a significant difference between the two years for each of the treatments (*p* < 0.001), suggesting that *phl*D abundance at the flowering stage was not stable in any crop. 

At the harvesting stage (Figure 1C), *phl*D gene abundance was relatively similar in both years (*p* > 0.05). However, there was also a significant difference in the *phl*D gene copy number between the crops (year-1, *p* < 0.05; and year-2, *p* < 0.005) and between the tillage practices (*p* < 0.05) in both of the seasons. The abundance of *phl*D was markedly higher under ST samples compared to CT in each crop. The OSR crop showed a significantly higher *phl*D copy number compared to the wheat crops (in rotation and monoculture). Furthermore, the presence of *phl*D was not detected in rotational wheat under CT (WCT) in either year. However, under strip tillage (ST), rotational wheat (WST) had a *phl*D abundance in the order of ~8 log per gram soil. Additionally, the wheat crop under monoculture (WC) exhibited a similar *phl*D gene abundance under both tillage regimes. 

### 3.2. Population Density of Indigenous phlD Microbes in the Roots of OSR and Wheat Crops under Different Tillage Systems

The *phl*D gene copy numbers were also quantified within the roots of crops at three plant growth stages in OSR and wheat (in rotation and monoculture) under CT and ST systems. The mean Ct value and associated calculated gene copy number per gram of each root sample are presented in Appendix A. 

In year-1 at the vegetative stage (Figure 2A), the abundance of *phl*D were significantly (*p* < 0.05) higher in the OSR roots under both tillage regimes (especially in the ST samples) compared to both wheat crops, whereas in year-2, the OST samples showed a reduction of ~0.5 log *phl*D gene copy numbers compared to year-1 OST samples. There were no other significant differences between the two years (*p* > 0.05). Both wheat crops (rotation and monoculture) under both tillage practices, and OSR under ST, showed similar levels of *phl*D copy numbers in both years.

At the flowering stage (Figure 2B), in general, there was a greater abundance of *phl*D in ST-treated plots than in the CT plots, but this only proved to be statistically significant for the WCT/WST plots in both years. WST was the only treatment that was statistically significantly different from the other treatments in year-1, due to a higher abundance of the *phl*D gene. This was also true for WST in year two. There was no significant difference (*p* > 0.05) between year-1 and year-2 for any of the crops at this stage. 

At the harvesting stage (Figure 2C), again the *phl*D copy number tended to be higher in the strip tillage plots. However, this was statistically significant in OSR (~2.5 to 3 log) under ST in comparison to CT samples (*p* < 0.005), and OST showed the highest *phl*D abundance compared to wheat under both tillage practices This was observed in both year-1 and year-2. In addition, *phl*D abundance was significantly greater in WCST compared to WCCT in year-1 but not in year-2. Moreover, wheat monoculture presented higher *phl*D^+^ counts under ST in comparison to CT in year-1, whereas in year-2, the abundance of the *phl*D microbes increased under CT and reached levels similar to the ST samples (at ~8.5 log *phl*D per gram of root). The wheat crop in rotation showed similar *phl*D^+^ abundance under both tillage regimes in both years. However, the abundance was lower under ST compared to the other two crops of ST samples.

### 3.3. Relationships between Microhabitat Zones and Crops of Antimicrobial Gene phlD Abundance at Different Plant Growth Stages in Two Continuous Years

The results obtained from quantification of the *phl*D gene in the rhizosphere and root of OSR, rotation wheat and monoculture wheat crops are displayed in a gradient map, based on the growth stages with subgroups of tillage and year (Figure 3). This graph shows three major clusters where the first cluster consisted of both wheat crops’ (in rotation and monoculture) rhizosphere samples, the second cluster consisted of both wheat crops’ root samples and the third cluster comprised OSR rhizosphere and root samples. The heatmap shows that wheat rotation rhizosphere samples at the vegetative stage, and OSR root rhizosphere samples at the vegetative and harvesting stages had the highest *phl*D bacterial abundance under ST. The vegetative stage showed higher *phl*D counts under both tillage practices in both microhabitat zones of wheat and OSR crops compared to other growth stages. Intriguingly, at the flowering stage under the CT system, all the crops under both microhabitat zones showed similar *phl*D gene copy numbers in year 1. However, year-2 in the flowering stage exhibited a remarkable decrease in *phl*D counts compared to year-1 in each crop’s microhabitat zone except for the roots of wheat rotation. At the harvesting stage under the ST regime, all crops and their microhabitat zones showed highly stable *phl*D bacterial abundance in both years. Interestingly, the *phl*D bacterial population was the highest in the OSR rhizosphere/root followed by the wheat monoculture root, wheat rotation root and then the rhizospheres of both wheat crops under ST. 

There was a clear trend identified in the rhizosphere of wheat rotation under ST, where the *phl*D microbial population was highest at the vegetative stage and subsequently decreased at the flowering stage followed by the harvesting stage. Likewise, a similar trend was observed in the CT regime with the maximum *phl*D gene copy number at the vegetative stage, reduced at flowering and not detected at the harvesting stage. Wheat monoculture showed a similar trend to that of wheat rhizosphere under the ST regime. However, under CT, the rhizosphere of wheat monoculture responded differently to that of wheat rotation, though the crop species and variety were the same. Likewise, the roots of both wheat crops under both tillage systems and all growth stages displayed different *phl*D gene copy numbers. Overall, based on the crop species, OSR showed higher *phl*D abundance compared to both wheat crops. Moreover, conservation strip tillage (ST) presented highly stable *phl*D counts in both years compared to conventional tillage (CT) at all growth stages.

## 4. Discussion

Take-all disease, caused by *Gaeumannomyces graminis* var. *tritici*, continues to be an important root disease of wheat worldwide [32] and is at its most severe when wheat is grown in areas with high precipitation such as Northwest Europe. Early infection often results in yellowing of lower leaves, stunting and the premature death of plants in patches. Methods of chemical control have had only moderate success in controlling the disease [32]. Crop rotation and tillage practices can help to manage take-all disease. Many wheat cultivars lack resistance to take-all, but when grown in rotation with other crops, they have been shown to experience lower disease incidence. Certain crop species have been shown to reduce the build-up of *G. graminis* var. *tritici* inoculum in soils, breaking the cycle of infection [33]. These ‘break crops’ refers to specific crops that are grown in rotation with other cereal crops, primarily to break the cycle of pests and diseases, or to improve soil health [34]. Numerous studies have shown that *Brassica napus* (OSR) is an excellent break crop for cereal-based production systems due to it breaking the disease cycle of cereal pathogens, biofumigation and high amounts of residual N as well as more efficient N recovery [35].

Soils with a high population of 2,4-DAPG-producing bacteria are considered to be disease-suppressive because they naturally protect plants from diseases. Agronomists and farmers often seek to enhance the presence of these beneficial bacteria in their soils to promote healthy plant growth and reduce the need for chemical pesticides [36]. The effectiveness of 2,4-DAPG in disease suppression can vary depending on various factors, including the specific pathogen involved, environmental conditions and the overall microbial community in the soil. There is abundant microbiological and biochemical evidence demonstrating 2,4-DAPG producing *Pseudomonas* spp. play a key role in the control of take-all disease in wheat in fields throughout the United States [37] and in the Netherlands [38]. Wheat cultivars and crop species differ in how well they support 2,4-DAPG-producing pseudomonads and 2,4-DAPG production. 2,4-DAPG producers naturally occur in wheat soil microbiomes at low densities. Numerous factors can affect the number of 2,4-DAPG-producing bacteria in the soil and plant environment. The influence of different agricultural practices, and in the selection and proliferation of beneficial microbes such as *Pseudomonas*spp. in plant-associated microhabitat zones remain under studied. Therefore, this current research was conducted to explore the combined impact of crop rotation (OSR–wheat crop rotation and wheat monoculture) along with tillage practices (conventional tillage and conservation strip tillage) on 2,4-DAPG-producing bacterial spp. in the rhizospheres and roots of OSR and wheat crops at different plant growth stages over a period of two consecutive years. Although bacterial genera other than *Pseudomonas* have also been shown to produce 2,4-DAPG (e.g., *Enterobacter*) [39,40], we make the assumption that all the *phl*D copies detected originated from *Pseudomonas*, as the primer set that we used was based on the *phl*D gene in *Pseudomonas* spp. Therefore, our data may show an overestimation of the *Pseudomonas* populations with 2,4-DAPG-producing potential. 

Our study showed that *phl*D^+^ abundance in both OSR and wheat roots and rhizospheres were in considerably high numbers, typically being log 5–log 8 g^−1^. This was relatively consistent across the three growth stages sampled. The exception to this was wheat under conventional tillage at the harvesting stage, where the *phl*D gene was not detected in any of the wheat rhizosphere soil samples. Castro et al. [41] found much lower *phl*D numbers in wheat, with an average of just 3.5 log. However, they inoculated wheat with a DAPG+ *P. protegens* strain that resulted in *phl*D+ numbers of log 5.28 CFU g^−1^ root in season 1, and from log 4.46 to 5.45 CFU g^−1^ in season 2. Typically, the occurrence of pseudomonads in the different roots/rhizospheres of crop plants ranges from log 4–log 8 [12]. Raaijmakers et al. [42] detected log 5.7–6.3 CFU g^−1^ soil in the rhizosphere of wheat grown in fields with soil suppressive or conductive to take-all. Patel and Archana [26] used qPCR detection of the *phl*D gene to successfully track the survival of a *Pseudomonas* strain, genetically engineered to produce 2,4-DAPG when used to inoculate rice plants. However, there are other mechanisms for detecting 2,4-DAPG-producing bacteria. Hansen et al. [43] recently constructed a whole cell biosensor for detecting 2,4-DAPG-producing *Pseudomonas*. They created a sensor module using the *phl*F gene and coupled it to a *LacZ-Lux* reported cassette. They used this biosensor to successfully identify 2,4-DAPG-producing bacteria in grassland soils. 

Picard and Bosco [3] suggested that alterations in the population, diversity and colonisation patterns of *phl*D+ microbes in rhizosphere and root habitats are determined by the crop species, varieties and growth stage. This variability over the growing season might be attributed to the development of root systems and associated changes in rhizodeposits [6]. We observed significantly greater *phl*D copy numbers in the OSR root compared to the wheat root only at the vegetative stage, and in the rhizosphere, only at the harvesting stage. However, our results found little variation in the numbers of *phlD* microbes due to the plant growth stage or plant species. Interestingly, the rotation of OSR with wheat did not result in any increase in *phlD* copy numbers, compared to continuous monoculture wheat. Therefore, the use of OSR as a break crop for reducing take-all decline must be down to some other mechanism, or perhaps simply due to a reduced pathogen load resulting from niche exclusion when OSR is in the field.

Vian et al. [44] reported that tillage practices bring physicochemical changes in soil that may change the microbial population in soil over time. The most pronounced effect on *phlD* gene copy number was due to the tillage method. Conservation strip tillage resulted in significant increases in *phlD* in rhizosphere soil. These increases were consistently observed regardless of plant species, stage of growth or year. Rotenberg et al. [2] reported that farm management practices can influence the overall structure of the microbial community in soil. Changes in soil populations might reasonably be expected to alter the rhizosphere community structure to some degree. Dennert et al. [45] used amplicon sequencing to demonstrate that *Pseudomonas* was among the most abundant bacterial taxa in the root microbiome of field-grown wheat. They also found that pseudomonads carrying the *phl*D gene were enriched in samples from farmed plots without tillage or with reduced tillage systems compared to intensively tilled soils. They also reported that disease resistance tests showed that soils from organic reduced tillage plots exhibited the highest disease resistance. In addition, Dennert et al. [45] utilized a whole cell 2,4-DAPG biosensor to report on *phl*D gene expression, but this did not reveal any differences in gene expression between soils from different cropping systems. Rotenberg et al. [2] also found that *phlD+* pseudomonads were more abundant in the rhizosphere of maize grown in no tillage plots compared to moderately tilled plots. This suggests that cropping systems with reduced tillage intensity can favour the abundance of *phlD+* pseudomonads in rhizosphere soil. The effect of tillage on the *phlD* gene copy number in the roots was less pronounced and was inconsistent across growth stages, plant species and tillage practice. This result is in agreement with the finding of Dennert et al. [45] who also could not detect differences in the abundance of *phlD+* pseudomonads on roots between conventional no till, organic reduced tillage and intensively tilled systems. Our results suggest that plant-beneficial pseudomonads can be favoured by certain soil tillage systems, but soil resistance against plant diseases is likely determined by a multitude of biotic factors in addition to *Pseudomonas*.

Mavrodi et al. [29] reported the frequencies of *phl*D copy numbers were higher in the wheat rhizospheres of irrigated fields compared to dryland agriculture fields. Fluorescent pseudomonads are sensitive to reduced moisture levels and drought. Since conservation tillage practices are known to help soils retain soil water and organic carbon [27], these conditions may favour pseudomonads’ survival and allow their populations to grow and persist compared to soils under conventional tillage systems.

## 5. Conclusions

The result of the qPCR detection of the *phl*D gene showed it to have very high abundances both in the rhizosphere soils and root tissues of *Brassica napus* and *Triticum aestivum,* regardless of growth stage, tillage system or crop rotation. The results do show that conservation strip tillage resulted in significantly higher gene copy numbers of *phl*D in the rhizosphere soils of both winter oilseed rape and wheat, regardless of crop rotation regime. Conservation tillage had less of an effect on the *phl*D gene copy numbers in the roots. While there was an overall trend of increased *phl*D gene copy numbers in root samples, it was only statistically significant in 50% of the root samples. 

Overall, crop rotation did not appear to lead to an increase in *phl*D gene abundance in wheat compared to continuous wheat cultivation. Therefore, our hypothesis that wheat grown in rotation with OSR would have a greater abundance of *phl*D+ *Pseudomonas* is not supported by the data. 

The global excessive use of chemical pesticides with harmful side effects to the environment and human health has led to increased calls for a pesticide-free agriculture. Biological control of plant pathogens by antagonistic micro-organisms may be a nature-based solution to help reduce chemical pesticide use. The rhizosphere holds the key to the next Green Revolution, whereby the development of innovative new crop varieties and management practices will allow plants to be far more capable of recruiting and utilizing beneficial microbes in the soil microbiome for growth promotion and disease control. These should also include plant-associated pseudomonads. Further research continues to explore how to optimize the use of 2,4-DAPG producers and other beneficial microbes to improve soil health and plant disease resistance.

## Figures and Tables

**Figure 1 microorganisms-11-02459-f001:**
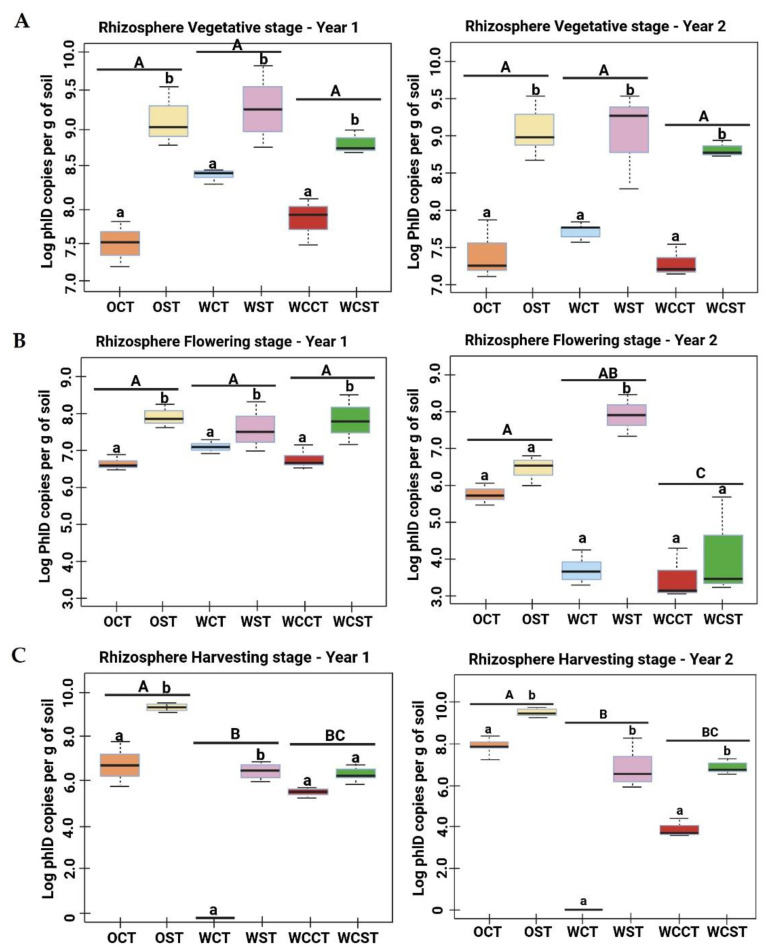
Population density of 2,4-DAPG-producing *Pseudomonas* in rhizosphere of OSR and wheat crops at different plant growth stages. Quantitative real-time PCR (qPCR) of *phl*D *Pseudomonas* populations in OSR and wheat rhizosphere, based on the number of *phl*D gene copies detected per gram of rhizosphere soil sample at three plant growth stages: (**A**) vegetative stage, (**B**) flowering stage, (**C**) harvesting stage under CT and ST in two continuous years of crop rotation. Uppercase letters denote statistically significant differences by Tukey post hoc tests, *p* < 0.05, between the crops. Lowercase letters denote statistically significant differences by Tukey post hoc tests, *p* < 0.05, between the tillage practices within one crop. The samples’ abbreviations are OSR (O), wheat in rotation (W), wheat continuous (WC), conventional tillage (CT), conservation strip tillage (ST), samples collected in year 2014-15 (Year-1) and in year 2015-16 (Year-2).

**Figure 2 microorganisms-11-02459-f002:**
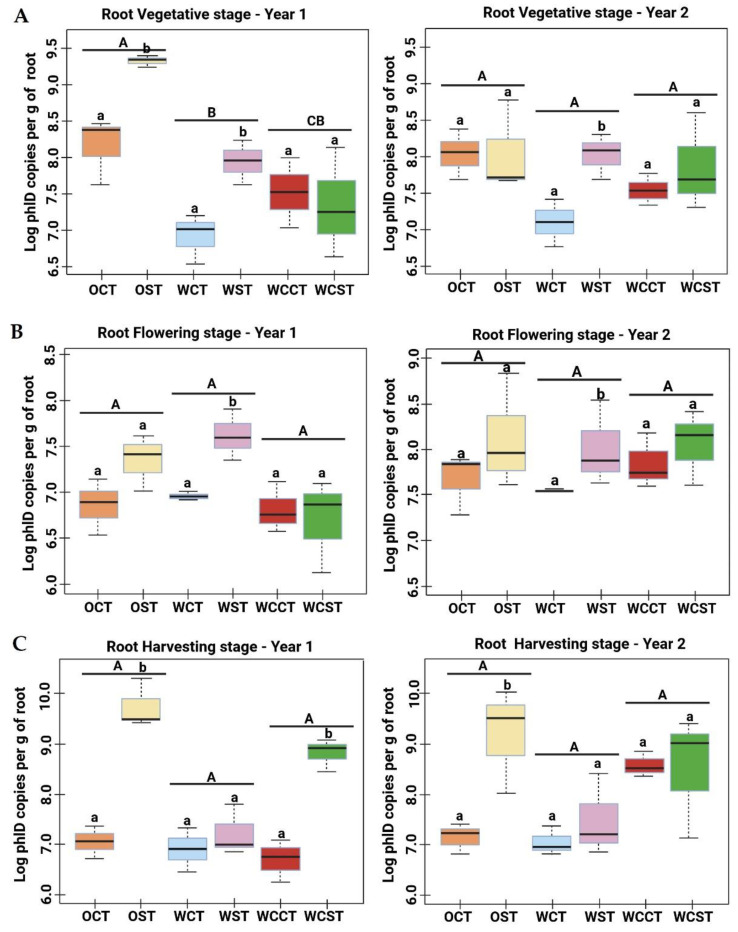
Population density of DAPG-producing *Pseudomonas* in roots of OSR and wheat crops at different plant growth stages. Quantitative real-time PCR (qPCR) of *phl*D *Pseudomonas* populations in OSR and wheat roots, based on the number of *phl*D gene copies detected per gram of root sample at three plant growth stages: (**A**) vegetative stage, (**B**) flowering stage, (**C**) harvesting stage under CT and ST in two continuous years of crop rotation. Uppercase letters denote statistically significant differences by Tukey post hoc tests, *p* < 0.05, between the crops. Lowercase letters denote statistically significant differences by Tukey post hoc tests, *p* < 0.05, between the tillage practices within one crop. The samples abbreviations are OSR (O), wheat in rotation (W), wheat continuous (WC), conventional tillage (CT), conservation strip tillage (ST), samples collected in year 2014-15 (Year-1) and in year 2015-16 (Year-2).

**Figure 3 microorganisms-11-02459-f003:**
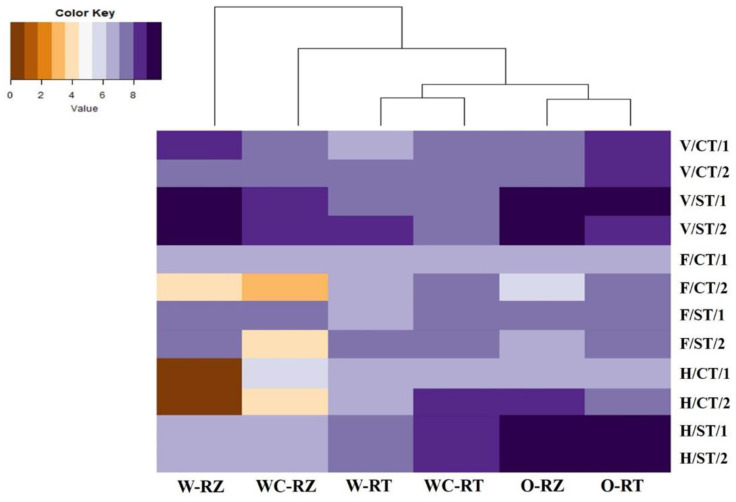
Heatmap showing normalized values of gene abundance in rhizosphere and root samples of OSR and wheat plants under CT and ST at three different growth stages. The colour scale depicts lowest (brown) through intermediate (white) to highest (violet) values for each variable. The individual symbols indicate OSR (O), wheat after OSR (W), wheat continuous (WC), vegetative stage (V), flowering stage (F), harvesting stage (H), conventional tillage (CT), conservation strip tillage (ST), rhizosphere (RZ), root (RT) samples collected in year 2014-15 (Year-1) and in year 2015-16 (Year-2).

**Table 1 microorganisms-11-02459-t001:** Primers used for *phlD* qPCR optimization.

Primer	Sequence (5′–3′)	Amplicon Length (bp)	Reference
B2BF	ACCCACCGCAGCATCGTTTATGAGC	319 bp	[23]
B2BR3	AGCAGAGCGACGAGAACTCCAGGGA		

## Data Availability

The data is available in the Appendix A.

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
