# Peer review of "Investigating the Impact of Tillage and Crop Rotation on the Prevalence of phlD-Carrying Pseudomonas Potentially Involved in Disease Suppression"

_microorganisms, 2023, doi:10.3390/microorganisms11102459_

Round 1
Reviewer 1 Report
Thank you for allowing me to review this manuscript on the effects of tillage and crop rotation on phlD-carrying Pseudomonas populations. Overall, this is an interesting topic but the manuscript would benefit from refining the framing, bolstering the methods details, providing more nuanced results/discussion, and highlighting concrete conclusions.
Here are some detailed comments and suggestions:
Major Revisions:
- The introduction provides good background but is missing some key information to set up the rationale and importance of the study. Please consider this paper, Residue retention and minimum tillage improve physical environment of the soil in croplands: A global meta-analysis.
- Expand on the role of phlD+ Pseudomonas in disease suppression and plant growth promotion.
- Highlight the knowledge gaps around how agricultural practices influence these beneficial microbes.
- Articulate the novelty/importance of evaluating combined effects of tillage and rotation.
- In the methods:
- Give more specifics on the crop management, fertilization, pest control etc. in the different rotation systems.
- Provide details on DNA extraction kits used.
- Explain how gene copy numbers were converted to population density.
- The results focus heavily on overall phlD abundance. It would be informative to also look at proportions of phlD+ populations.
- The discussion is underdeveloped. Expand on interpreting the key findings and practical implications for agronomic practices to promote disease suppressive microbial communities.
- Conclusions should highlight the main takeaways rather than just stating the aim. Add depth.
Minor Issues:
- Reduce repetition in the introduction and improve flow between paragraphs.
- Define abbreviations like CT, ST etc. at first use.
- Carefully proofread - there are multiple typos, grammar issues.
- Figures are low resolution and axis labels are unclear. Should be improved.
Author Response
Major Revisions:
- The introduction provides good background but is missing some key information to set up the rationale and importance of the study. Please consider this paper, Residue retention and minimum tillage improve physical environment of the soil in croplands: A global meta-analysis.
- Expand on the role of phlD+ Pseudomonas in disease suppression and plant growth promotion.
Additional information is provided in the introduction section
- Highlight the knowledge gaps around how agricultural practices influence these beneficial microbes.
The knowledge gaps are highlighted at the end of the introduction
- Articulate the novelty/importance of evaluating combined effects of tillage and rotation.
- In the methods:
- Give more specifics on the crop management, fertilization, pest control etc. in the different rotation systems.
These are provided in the supplementary file Table S2.
- Provide details on DNA extraction kits used.
The kits used in the DNA extraction are detailed in the methods section
- Explain how gene copy numbers were converted to population density.
This information is now provided in the supplementary section S8.
- The results focus heavily on overall phlD abundance. It would be informative to also look at proportions of phlD+ populations.
This would be useful, however, unfortunately we didn’t collect this data.
- The discussion is underdeveloped. Expand on interpreting the key findings and practical implications for agronomic practices to promote disease suppressive microbial communities.
The discussion section has been extensively rewritten focusing on the main findings of the research
- Conclusions should highlight the main takeaways rather than just stating the aim. Add depth.
A conclusion section has now been included highlighting the main finding of the research.
Minor Issues:
- Reduce repetition in the introduction and improve flow between paragraphs.
- Define abbreviations like CT, ST etc. at first use.
These have now been included where first used.
- Carefully proofread - there are multiple typos, grammar issues.
The manuscript has been carefully checked and any typos and grammar issues have been corrected.
- Figures are low resolution and axis labels are unclear. Should be improved.
High resolution figures are now provided.

Reviewer 2 Report
The paper deals with comparison of the phID gene abundance (responsible to antifungal activity) in the rhizosphere and roots of rape and wheat under different growth conditions in sandy clay loam soil for two years. The topic is relevant and interesting. The paper is well written, with clear objectives. The methods used are appropriate. Authors collected and summarized a large amount of data. Overall, the structure of this paper is good. Some of the comments and suggestions are as follows:
TITLE. Please check the title. Only Brassica napus is mentioned. Wheat is missing.
Abstract. The Abstract section needs more specific data. The statement on mechanisms (L12-14) can be omitted because authors did not investigate the mechanisms of bacterial antagonism with fungi by means of phID gene. Instead, it is recommended to mention the main findings summarized in Figure 3, L319-320.
L391-397. Discussion on the effect of moisture on plant colonisation frequencies of phlD copy number in rhizosphere: there is a lack of experimental data. Table S4 contains a lot of data related to average levels of temperature/humidity/wind speed, but irrespectively of CT vs. ST. So this assumption (L396-397) sounds quite speculative. It is suggested to discus the differences of soil physicochemical properties which were supported by measurements.
Please discuss why the lowest gene abundance was detected at the flowering stage.
Please discuss possible effects of pesticide application (Table S2) on the abundance of phID gene in rhizosphere.
L469-470. This statement contradicts with that on L371-376.
L222. “…under Convention Tillage (CT) and Strip Tillage (ST)…” It is not necessary to describe abbreviations CT and ST again. It has been described above (L65).
Table S1. TYPO in the footnotes (“…were highted with orange colour….”).
L410. TYPO “…that the environmental condition was very in flowering stage in both years which…”
Kind regards
Author Response
TITLE. Please check the title. Only Brassica napus is mentioned. Wheat is missing.
The title has now been shortened and no longer refers to either of the plant species and focuses just on tillage and crop rotation.
Abstract. The Abstract section needs more specific data. The statement on mechanisms (L12-14) can be omitted because authors did not investigate the mechanisms of bacterial antagonism with fungi by means of phID gene. Instead, it is recommended to mention the main findings summarized in Figure 3, L319-320.
The abstract is updated to better reflect the results of the study.
L391-397. Discussion on the effect of moisture on plant colonisation frequencies of phlD copy number in rhizosphere: there is a lack of experimental data. Table S4 contains a lot of data related to average levels of temperature/humidity/wind speed, but irrespectively of CT vs. ST. So this assumption (L396-397) sounds quite speculative. It is suggested to discus the differences of soil physicochemical properties which were supported by measurements.
The discussion has been significantly revised and we now discuss the possible influence of variations in the soil physicochemical properties which we recorded.
Please discuss why the lowest gene abundance was detected at the flowering stage.
This is now discussed.
Please discuss possible effects of pesticide application (Table S2) on the abundance of phID gene in rhizosphere.
It is unclear why this fungicide would affect phlD/2,4-DAPG+ pseudomonads in the soil. We have no data to support any possible effects.
L469-470. This statement contradicts with that on L371-376.
This comment is unclear- L371 refer to Pseudomonads as not being the only possible harborers of the 2,4-DAPG phenotype. Whereas L469 refers to the fact that mechanisms other than DAPG are likely to be involved in disease suppression.
L222. “…under Convention Tillage (CT) and Strip Tillage (ST)…” It is not necessary to describe abbreviations CT and ST again. It has been described above (L65).
Agreed this has been changed.
Table S1. TYPO in the footnotes (“…were highted with orange colour….”).
This has been corrected
L410. TYPO “…that the environmental condition was very in flowering stage in both years which…”
This has been corrected.
